# Impact of Anti-SARS-CoV-2 Vaccination on Disease Severity and Clinical Outcomes of Individuals Hospitalized for COVID-19 Throughout Successive Pandemic Waves: Data from an Italian Reference Hospital

**DOI:** 10.3390/vaccines12091018

**Published:** 2024-09-06

**Authors:** Annalisa Mondi, Ilaria Mastrorosa, Assunta Navarra, Claudia Cimaglia, Carmela Pinnetti, Valentina Mazzotta, Alessandro Agresta, Angela Corpolongo, Alberto Zolezzi, Samir Al Moghazi, Laura Loiacono, Maria Grazia Bocci, Giulia Matusali, Alberto D’Annunzio, Paola Gallì, Fabrizio Maggi, Francesco Vairo, Enrico Girardi, Andrea Antinori

**Affiliations:** 1Clinical and Research Department, National Institute for Infectious Diseases Lazzaro Spallanzani IRCCS, 00149 Rome, Italy; annalisa.mondi@inmi.it (A.M.); carmela.pinnetti@inmi.it (C.P.); valentina.mazzotta@inmi.it (V.M.); angela.corpolongo@inmi.it (A.C.); alberto.zolezzi@inmi.it (A.Z.); samir.almoghazi@inmi.it (S.A.M.); laura.loiacono@inmi.it (L.L.); mariagrazia.bocci@inmi.it (M.G.B.); andrea.antinori@inmi.it (A.A.); 2Department of Epidemiology, National Institute for Infectious Diseases Lazzaro Spallanzani IRCCS, 00149 Rome, Italy; assunta.navarra@inmi.it (A.N.); claudia.cimaglia@inmi.it (C.C.); alessandro.agresta@inmi.it (A.A.); francesco.vairo@inmi.it (F.V.); 3Laboratory of Virology, National Institute for Infectious Diseases Lazzaro Spallanzani IRCCS, 00149 Rome, Italy; giulia.matusali@inmi.it (G.M.); fabrizio.maggi@inmi.it (F.M.); 4Department of Life, Health and Environmental Sciences, University of L’Aquila, 67100 L’Aquila, Italy; 5Health Direction, National Institute for Infectious Diseases Lazzaro Spallanzani IRCCS, 00149 Rome, Italy; paola.galli@inmi.it; 6Scientific Direction, National Institute for Infectious Diseases Lazzaro Spallanzani IRCCS, 00149 Rome, Italy; enrico.girardi@inmi.it

**Keywords:** COVID-19, hospitalization, critical illness, death, COVID-19 vaccines, COVID-19 vaccine booster shot

## Abstract

This is a retrospective observational study including all COVID-19 patients admitted at our Institute throughout three successive pandemic waves, from January 2021 to June 2023. The main in-hospital outcomes (clinical progression [CP], defined as admission to Intensive Care Unit [ICU]/death, and death within 28 days) were compared among participants unvaccinated (NV), fully vaccinated (FV), with one (FV&B1) and two (FV&B2) booster doses. Vaccinated participants were stratified into recently and waned FV/FV&B1/FV&B2, depending on the time elapsed from last dose (≤ and >120 days, respectively). There were 4488 participants: 2224 NV, 674 FV, 1207 FV&B1, and 383 FV&B2. Within 28 days, there were 604 ICU admissions, 396 deaths, and 737 CP. After adjusting for the main confounders, the risk of both in-hospital outcomes was reduced in vaccinated individuals, especially in those who received the booster dose (approximately by 36% for FV and >50% for FV&B1 and FV&B2 compared to NV). Similarly, after restricting the analysis to vaccinated participants only, we observed a risk reduction of approximately 40% for FV&B1 and 50% for FV&B2, compared to FV, regardless of the distance since the last dose. Our data confirm the vaccine’s effectiveness in preventing severe COVID-19 and support the efforts to increase the uptake of booster doses, mainly among older and frailer individuals, still at a greater risk of clinical progression.

## 1. Introduction

Since SARS-CoV-2 emerged as a new respiratory virus responsible for more than 770 million cases of confirmed COVID-19 so far, including about 7 million deaths worldwide [1], many efforts have been made to control its spread. Among them, the development of vaccines against SARS-CoV-2 at an unprecedented rapid pace and the launch of a mass vaccination campaign have been the key strategy to control the spread of the virus and prevent COVID-19-related hospitalizations and deaths. In Italy, the COVID-19 vaccination campaign started on 27 December 2020 with a frailty- and age-stratified roll-out approach [2]. In this first phase, four types of vaccines were progressively available in Italy: BNT162b2, mRNA-1273, ChAdOx1, and Ad26.COV2.S. By September 2021, the administration of the first booster dose of COVID-19 vaccine was approved by the Italian Ministry of Health for those who completed the first vaccination schedule [3]. The administration of the second booster dose was approved in February 2022 initially for immunocompromised subjects and subsequently extended to the entire population [4]. Booster doses consisted of BNT162b2 or mRNA-1273 vaccines.

COVID-19 vaccines demonstrated a high effectiveness in reducing the risk of SARS-CoV-2 infections and the burden of COVID-19 severe disease and mortality in both clinical trials [5,6] and in the real-world setting [7,8]. Nevertheless, symptomatic SARS-CoV-2 infections still occur in fully vaccinated subjects. The risk of developing vaccine breakthrough infections is related to several factors as host determinants (e.g., age, comorbidities, and immunological status), vaccine-related factors (e.g., dose number, and time elapsed since the last dose), and viral characteristics (e.g., viral variant of concern [VoC]) [9]. In particular, there is now clear evidence that vaccine effectiveness weans over time and the administration of a booster dose is highly effective in restoring good protection against symptomatic infection and severe disease [10,11,12,13,14,15,16]. In addition, concerning the viral variant, a lower vaccine effectiveness has been reported for Omicron compared to previous VoCs, probably related to the greater immune-evasion capacity of this variant [9,17].

Although vaccine breakthrough infections are generally milder and carry a lower risk of hospitalization, the risk of severe COVID-19 remains, particularly among groups at a higher risk of severe disease [9,18]. Vaccinated individuals who require hospitalization for severe SARS-CoV-2 infections seemed to have a lower risk of clinical progression and fatal outcomes [18,19,20,21,22,23,24]. However, limited data on the impact of vaccination on hospitalized individuals are available, especially in the setting of Omicron dominance and in fully vaccinated and boosted subjects.

The aim of this study was to compare clinical characteristics and in-hospital outcomes of patients requiring hospitalization for SARS-CoV-2 infection according to vaccination status over a 30-month observation period starting from the date of the introduction of the anti-SARS-CoV-2 vaccine campaign in Italy.

## 2. Materials and Methods

### 2.1. Study Design and Population

We conducted a retrospective observational monocenter study including all adult subjects who were admitted to the National Institute for Infectious Diseases (INMI) “Lazzaro Spallanzani”, in Rome, Italy, between 1 January 2021 to 30 June 2023 with a discharge diagnosis of SARS-CoV-2 infection, confirmed by real-time polymerase chain reaction (RT-PCR) on at least one respiratory specimen, and available complete data on anti-SARS-CoV-2 vaccination from the regional vaccination register (Anagrafe Vaccinale Regione Lazio). According to vaccination status at hospital admission, participants were divided into four exposure groups: 1. Not vaccinated (NV), if they did not receive any vaccine dose, or they received only the first dose of a 2-dose vaccine series less than 14 days before the hospital admission; 2. Fully vaccinated (FV), if they completed the vaccine schedule more than 14 days before the hospital admission; 3. Fully vaccinated and one booster dose (FV&B1), if they completed the primary vaccine schedule and received the first booster dose of vaccine at least 14 days before the hospital admission; and 4. Fully vaccinated and two booster doses (FV&B2), if they completed the primary vaccine schedule and received the second booster dose of vaccine at least 14 days before the hospital admission. Subjects who had received partial vaccination (defined as only the first dose of a 2-dose vaccine series or the complete primary vaccine schedule less than 14 days before the hospital admission) were excluded from the analysis. Likewise, individuals who were not resident in the Lazio region and, therefore, had no vaccination data available from the regional vaccine registry were not included in the present study. Finally, we also excluded patients admitted within 90 days from previous hospitalization, assuming they had a prolonged infection. Demographic and clinical data including age, sex, country of birth, number and type of comorbidities (detailed in supplementary methods), characteristics of COVID-19 (date of symptoms onset, presence of pneumonia, and disease severity) were extracted from hospital discharge records and COVID-19 notification forms. Laboratory and virologic data were taken from hospital laboratory electronic database. Moreover, information about previous SARS-CoV-2 infections were obtained through RT-PCR tests collected by Lazio regional integrated surveillance system platform, COVID-19 notification forms, and results of SARS-CoV-2 serology (particularly Immunoglobulin G [IgG] anti-receptor-binding domain [RBD] of the spike (S) protein) performed at hospital admission for unvaccinated patients. Finally, pandemic waves were defined based on the predominant viral variant in circulation, according to the COVID-19 national surveillance data produced by the Italian National Institute of Health [25] and referring to previously published data from our hospital [26]. Follow-up accrued from the date of hospital admission (study baseline) to the achievement of the defined clinical outcomes (clinical progression or in-hospital death) or to day 28 of hospitalization or to hospital discharge, whichever came first.

### 2.2. Statistical Analysis

Descriptive characteristics at hospital admission were provided using median values and interquartile ranges (IQR) for continuous variables, and frequencies and percentages for categorical variables, were reported by vaccination status (NV, FV, FV&B1, and FV&B2) and were compared by the four groups using Chi-square test (Fisher’s exact test when applicable) for categorical variables and Kruskal–Wallis test for continuous variables. Similarly, comparison of main clinical and virological outcomes was assessed according to vaccination status. Predictive factors of clinical progression and in-hospital death within 28 days from hospital admission were assessed using a multivariable logistic regression calculating odds ratios (MLR-OR) and their 95% confidence intervals (95% CI), both in total population and after restricting the analysis only to vaccinated population (FV, FV&B1, and FV&B2). To include all potential explanatory variables each factor available at baseline was considered as a covariate in the MLR models. Age was assessed both in its continuous form and categorized in four groups (18–39, 40–59, 60–79, and ≥80); the form included in the logistic models was that which minimized the Bayesian information criterion (BIC). Multicollinearity between predictors was assessed using variance inflation factors (VIF) to exclude the presence of highly correlated variables that merit further investigation (it is generally accepted that a variable with VIF exceeding 10 indicates concerning collinearity) [27,28]. Clinical progression was defined as in-hospital death for any cause or admission to intensive care unit (ICU) within 28 days from hospital admission. In addition, the distribution of the four exposure groups in the study population, only represented by patients hospitalized for COVID-19, was described in comparison with the levels of vaccination coverage observed in the regional population, for each trimester of the study period, both overall and by age group. The data on vaccinations coverage in the Lazio Region were extracted from the open data repository [29]. A two-tailed *p*-value less than 0.05 indicated conventional statistical significance. All statistical analyses were performed using STATA release 17 (StataCorp. 2021. StataCorp LLC, College Station, TX, USA), while the forestploter R package, version 1.1.2, was utilized for the visual representation of MLR-ORs and their 95% CIs [https://CRAN.R-project.org/package=forestploter, accessed on 16 August 2024].

## 3. Results

Over an observation period of 30 months, 4950 patients with a diagnosis of COVID-19 were admitted at our Institute. Of these, 4488 were included in the study: 2224 (49.6%) NV, 674 (15.0%) FV, 1207 (26.9%) FV&B1, and 383 (8.5%) FV&B2. The details of the patients’ selection are depicted in Figure 1.

### 3.1. Patients’ Characteristics at Hospital Admission

The main characteristics at hospital admission according to vaccination status are shown in Table 1.

Briefly, the study population consisted mostly of men (59.4%), of Italian origin (88.4%), with a median age of 68 (IQR 54–80) years, and with at least one concomitant disease (71.1%). At hospital admission, which occurred within a median time of 4 days (IQR 2–8) from the symptoms’ onset, 89.9% of patients had evidence of pneumonia. The four exposure groups differed for most of the main baseline characteristics. In particular, vaccinated subjects were more likely to be older (FV&B2 82 [IQR 74–87] years, FV&B1 76 [IQR 64–84] years, FV 68 [IQR 55–79] years, and NV 59 [IQR 48–72] years, *p* < 0.001) and Italian (FV&B2 98.4% vs. FV&B1 93.6% vs. FV 89.3%, and NV 83.6%, *p* < 0.001), to be admitted after a shorter median time from symptoms’ onset (FV&B2 2 [IQR 1–4] days, FV&B1 2 [IQR 1–5] days, FV 4 [IQR 2–7] days, and NV 6 [IQR 3–9] days, *p* < 0.001), and to have multiple comorbidities (patients with more than one comorbidity: FV&B2 53.9 vs. FV&B1 42.0% vs. FV 40.1%, and NV 25.2%, *p* < 0.001), as compared to NV ones. In all the exposure groups, cardiovascular disease was the most common concomitant illness (FV&B2 69.5% vs. FV&B1 57.2% vs. FV 53.6%, and NV 43.8%, *p* < 0.001), followed by chronic respiratory disease (FV&B2 31.9% vs. FV&B1 26.6% vs. FV 21.7%, and NV 15.5%, *p* < 0.001) and diabetes (FV&B2 24.8% vs. FV&B1 16.6% vs. FV 20.0%, and NV 12.6%, *p* < 0.001). Almost all reported comorbidities, except metabolic disease (including obesity and dyslipidemia), were more frequent among vaccinated patients than not vaccinated ones. At the time of hospital admission, NV patients compared to the other exposure groups were more likely to have lung involvement at the CT scan (NV 95.2% vs. FV 89.0% vs. FV&B1 80.9% vs. FV&B2 88.5%, *p* < 0.001). Among vaccinated patients, mRNA vaccine BNT162b2 was the most frequent type of vaccine administered for the first dose in the FV, FV&B1, and FV&B2 groups (66.3%, 78.3%, and 81.7%, respectively, *p* < 0.001). Of note, a total of 441 (19.5%) subjects (16.8% FV, 18.2% FV&B1, and 28.2% FV&B2) received their last dose within 120 days from hospital admission.

### 3.2. Vaccination Coverage over Time in the Study Population and in the Lazio Region

The prevalence of subjects who received a full primary cycle of anti-SARS-CoV-2 vaccination before hospital admission increased gradually over time, from none of the 197 cases hospitalized in January 2021 to 47% of those (n = 250) hospitalized in November 2021, two months after the approval of a booster dose in Italy. The proportion of subjects who received the additional dose of vaccine after the primary cycle (FV&B1) progressively raised, ranging from around 1% of the patients admitted for COVID-19 in November 2021 (n = 250) to 77% in June 2022 (n = 148); similarly, an increasing proportion of individuals who received a second booster dose was observed from February to December 2022 (from less than 1% [1/139] to 30% [56/184]). Thereafter, the proportions remained roughly stable (Figure 2).

The temporal change in anti-SARS-CoV-2 vaccination coverage described in our study population over the observation period is consistent with the regional data. Of note, the proportion of NV was higher in our population than in the general population across all age strata. Similarly, the proportion of FV&B2 was higher among our study population compared to the regional data in the strata from 20 to 69 years (Appendix A).

### 3.3. Main Clinical and Virological Outcomes

Over a median follow-up of 14 days (IQR 9–22), 396 subjects (8.8%) died, 604 (13.5%) were admitted to ICU, and 737 (16.4%) experienced clinical progression within 28 days from hospital admission. Looking at the comparisons among crude rates, we did not observe any significant differences in terms of 28-day in-hospital death by vaccination status (NV 8.3% vs. FV 9.2% vs. FV&B1 9.5% vs. FV&B2 9.1%, *p* = 0.627). Conversely, a higher proportion of NV participants compared to vaccinated ones was admitted to the ICU (NV 16.4% vs. FV 11.7% vs. FV&B1 10.1% vs. FV&B2 10.2%, *p* < 0.001) and experienced clinical progression (NV 18.2% vs. FV 15.6% vs. FV&B1 14.3% vs. FV&B2 14.4%, *p* = 0.012) within 28 days from hospitalization. The median length of hospital stay and ICU stay did not significantly differ among the exposure groups (Table 2). Concerning virological outcomes, a higher proportion of individuals with persistent viral shedding within day 28 of hospitalization was observed among those who received booster doses (NV 83.2% vs. FV 82.4% vs. FV&B1 89.2% vs. FV&B2 92.2%, *p* < 0.001) (Table 2).

### 3.4. Predictors of 28-Day in-Hospital Death and Clinical Progression

Looking to the predictive factors for the main in-hospital outcomes, after adjusting for the main confounders listed in the methods section, having received a complete primary cycle of anti-SARS-CoV-2 vaccination compared to being not vaccinated was associated with a 36% (MLR-OR FV vs. NV 0.64, 95% CI 0.45–0.90, *p* = 0.011) and 43% (MLR-OR FV vs. NV 0.57, 95% CI 0.44–0.75, *p* < 0.001) reduction in the risk of 28-day in-hospital death and clinical progression, respectively. Similarly, having received booster doses versus not being vaccinated reduced the risk for both outcomes by more than 50% (MLR-OR FV&B1 vs. NV 0.48, 95% CI 0.35–0.65, *p* < 0.001, and MLR-OR FV&B2 vs. NV 0.33, 95% CI 0.21–0.51, *p* < 0.001 for in-hospital death; MLR-OR FV&B1 vs. NV 0.47, 95% CI 0.37–0.61, *p* < 0.001, and MLR-OR FV&B2 vs. NV 0.39, 95% CI 0.27–0.55, *p* < 0.001 for clinical progression). The female gender was also associated to a lower risk of both 28-day in-hospital death (MLR-OR 0.63, 95% CI 0.50–0.80, *p* < 0.001) and clinical progression (MLR-OR 0.67, 95% CI 0.56–0.80, *p* < 0.001). On the contrary, older age (MLR-OR per 10 years older, for in-hospital death, 1.63, 95% CI 1.48–1.80, *p* < 0.001 and, for clinical progression, 1.20, 95% CI 1.12–1.28, *p* < 0.001) and certain underlying comorbidities such as cardiovascular disease (MLR-OR, for in-hospital death, 1.43, 95% CI 1.12–1.84, *p* = 0.005 and, for clinical progression, 1.64, 95% CI 1.36–1.98, *p* < 0.001), metabolic disease (MLR-OR, for in-hospital death, 1.51, 95% CI 1.06–2.14, *p* = 0.021 and, for clinical progression, 1.46, 95% CI 1.13–1.89, *p* = 0.004), renal diseases (MLR-OR, for in-hospital death, 2.57, 95% CI 1.92–3.44, *p* < 0.001 and, for clinical progression, 2.16, 95% CI 1.66–2.80, *p* < 0.001), and neoplasms/hematologic diseases (MLR-OR, for in-hospital death, 3.09, 95% CI 2.23–4.29, *p* < 0.001 and, for clinical progression, 2.16, 95% CI 1.62–2.89, *p* < 0.001) showed a significant higher risk for both in-hospital outcomes; having an immunodeficiency status was associated with a higher risk only for in-hospital death (MLR-OR 2.65, 95% CI 1.53–4.58, *p* < 0.001). Finally, no association between pandemic waves and clinical outcomes for both in-hospital outcomes was observed (Figure 3a,b and Appendix A).

After restricting the analysis to the vaccinated population having received an immunization booster, compared with having completed only the first vaccination cycle, regardless of the time since the last vaccine dose, significantly decreased the risk of 28-day clinical progression and mortality (MLR-OR 0.63 for FV&B1, 95% CI 0.45–0.89, *p* = 0.008 and MLR-OR 0.52 for FV&B2, 95% CI 0.34–0.81, *p* = 0.004 vs. MLR-OR 0.61 for FV&B1, 95% CI 0.40–0.93, *p* = 0.021 and MLR-OR 0.45 for FV&B2, 95% CI 0.26–0.77, *p* = 0.004, respectively). In contrast, the time elapsed from the last dose did not show a significant effect on the risk of 28-day clinical progression and mortality. Similarly to the total population, in vaccinated subjects, older age (MLR-OR per 10 years increase for clinical progression 1.14, 95% CI 1.02–1.27, *p* = 0.019 and MLR-OR per 10 years increase for in-hospital death 1.52, 95% CI 1.30–1.76, *p* < 0.001) and having hematologic/neoplastic diseases (MLR-OR for clinical progression 2.37, 95% CI 1.70–3.31, *p* < 0.001 and MLR-OR for in-hospital death 3.53, 95% CI 2.42–5.17, *p* < 0.001), and renal impairment (MLR-OR for clinical progression 2.22, 95% CI 1.60–3.07, *p* < 0.001 and MLR-OR for in-hospital death 2.46, 95% CI 1.71–3.54, *p* < 0.001) were associated with a higher risk of both 28-day clinical outcomes. In addition, having an immunodeficiency status increased the risk of in-hospital mortality (MLR-OR for in-hospital death 2.37, 95% CI 1.22–4.63, *p* = 0.011). Finally, the female gender was confirmed to be associated with a reduced risk of clinical progression and in-hospital death also in the vaccinated population (MLR-OR 0.69, 95% CI 0.54–0.89, *p* = 0.004 vs. MLR-OR 0.67, 95% CI 0.49–0.91, *p* = 0.011, respectively) (Figure 4a,b and Appendix A).

## 4. Discussion

In this retrospective observational study, including 4488 subjects who were hospitalized at INMI “Lazzaro Spallanzani” between January 2021 and June 2023, the efficacy of the anti-SARS-CoV-2 vaccination, and particularly of booster doses, in reducing the risk of severe COVID-19 was confirmed over a considerable observation period including several pandemic waves.

Indeed, in our analysis, unvaccinated participants were more prone to developing critical disease as shown by the lower rate of ICU admission and clinical progression (in-hospital death and ICU admission within 28 days from hospital admission) among vaccinated individuals. Importantly, this result was more evident for boosted participants, underlying the importance of promoting access to additional doses mainly for frailer subjects. Conversely, when compared crude rates, we did not observe any difference by vaccination status in terms of in-hospital mortality for any causes, however lower (nearly 9% in the overall study population) compared to other reports in similar settings [19,23], and length of hospital and ICU stay, probably because other unidentified confounders may play a causative role and several complications may characterize the hospital stay of a population of old subjects underlying several medical conditions. Nevertheless, our adjusted analysis showed an increasing protection from in-hospital death and clinical progression as the number of vaccine doses received increases. In particular, we observed a risk reduction of more than 50% for boosted individuals compared to unvaccinated ones and, after restricting the analysis to vaccinated participants, we demonstrated a similar risk reduction for those who have received additional doses after the full primary cycle. This result is in line with the existing literature which demonstrates the association of vaccination status and additional doses with in-hospital death or ICU admission, among individuals hospitalized for COVID-19 [18,19,20,21,22,23,24]. Moreover, this study underlines the accuracy and robustness of some previous molecular studies indicating the efficacy of booster-induced memory B cells against Omicron and its variants at the atomic, cellular, and animal levels [30,31,32].

As expected, older participants and those with concomitant medical conditions demonstrated a heightened vulnerability to clinical progression and mortality, regardless of the number of doses received. Interestingly, among comorbidities, the greatest risk of developing critical COVID-19 and, in particular, in-hospital death was observed in the case of underlying renal disease or any immunocompromised status, due to an active treatment and/or immunocompromising medical condition such as immunodeficiency diseases and solid or hematologic neoplasms. The association was even stronger after considering only our vaccinated population. Increased clinical risk for older people has been already demonstrated [11,33] as a consequence of underlying frailty, comorbidity, and immune senescence. Similarly, patients with a chronic renal disease, in particular, those who are under dialysis or kidney transplant recipients, have been known to be at an increased risk of severe infection with SARS-CoV-2, and to have an impaired response to standard vaccination [34]. Finally, immunocompromised subjects are less likely to mount an adequate immune response to vaccination and more likely to develop severe COVID-19 and may have persistent clinical complications and prolonged SARS-CoV-2 positivity in respiratory samples [35]. Indeed, we observed a prolonged viral shedding among individuals vaccinated with booster doses, which is not surprising considering that they consist mainly of elderly, frail, and immunocompromised individuals who are less likely to shed the virus from respiratory samples. Of note, we did not find any association between pandemic waves and in-hospital clinical outcomes, underlying the importance of vaccination for specific categories of patients even with the more recent and less pathogenic SARS-CoV-2 variant [26,36].

In the present study, the time elapsing from the last dose was not associated with the risk of developing critical illness or mortality. Our result is apparently in contrast with the evidence arising from a recent report comparing the effectiveness of several booster doses with that of full primary vaccination received from April 2022 to July 2023 [15]. In fact, this study confirmed the importance of booster doses in restoring individual protection against COVID-19-related hospitalization and death and demonstrated the rapid decline in vaccine efficacy after 12 weeks, especially among individuals aged ≥ 80 years. However, our study was conducted over a larger observation period, covering a range of viral variants, and in a different setting, including only hospitalized patients of different ages, half of whom were not vaccinated. Indeed, other analyses from the US during 2022–2023 showed a similar waning of COVID-19 vaccine effectiveness against hospitalization, but also demonstrated a more sustained effectiveness against critical illness, with protection lasting well over 1 year after the most recent dose. Furthermore, other studies [37,38] found a waning of protection against COVID-19 in elderly patients after six months following the second vaccine dose, while, in the current analysis, we evaluated a distance of 120 days from the last dose. It is, therefore, clear that, particularly after hospitalization, a number of factors in addition to vaccination status should be considered when assessing key clinical outcomes, such as ICU admission or mortality, especially in a more medically fragile population, and further studies are needed to identify the real association between the waning of immunity after vaccination and the declining protection from severe COVID-19.

Finally, given the extensive observation period of our study, which spans from the beginning of the anti-SARS-CoV-2 vaccination campaign in Italy to June 2023, when the second booster dose had been available for over a year, we observed an increasing vaccination coverage, which reflects the regional data on vaccination uptake. Interestingly, by the end of 2022, the proportion of subjects who had received a second booster remained almost stable at a low level both in the study population and at the regional level, particularly among the youngest, indicating that, despite public initiatives promoting the administration of booster doses, vaccine hesitancy regarding additional doses persists, especially for those individuals who have a self-perception of being at a low risk of severe disease [39]. As expected, the proportion of unvaccinated patients was persistently higher in the study population, confirming the effectiveness of vaccination against hospitalization for COVID-19. In addition, it is worth noting that, the proportions of study participants with a second booster dose, compared with regional data, were similar or even higher, reflecting a high vaccination coverage mainly among those more vulnerable due to concomitant diseases and being at high risk of being hospitalized, regardless of age.

To our knowledge, this is the largest study ever conducted in the target population of individuals hospitalized for COVID-19 covering an observation period with a considerable length when several VoCs were circulating, allowing the inclusion of Omicron variants, associated to a less severe presentation of the disease [26,36], and boosted individuals. However, several limitations of our analysis need to be mentioned. First, the observational nature of the study is prone to residual and potential unmeasured confounding bias. Second, the monocentric design of the study, on the one hand, limits the generalizability of our findings, but, on the other hand, ensures homogeneity in background care, disease management, and resource availability, even in the absence of information on therapeutic approaches had during the hospitalization and any previous early treatment received. Finally, there is a lack of SARS-CoV-2 sequencing data to have a more detailed and precise definition of pandemic periods.

## 5. Conclusions

In conclusion, the present study confirms the effectiveness of complete anti-SARS-CoV-2 vaccination and, in particular, booster doses in reducing the risk of severe clinical progression and in-hospital death among individuals hospitalized for COVID-19. Indeed, unvaccinated participants were more prone to develop critical disease. In addition, our findings indicate a range of demographic and clinical factors, such as older age and the presence of certain concomitant medical conditions, associated with an increased clinical risk of severe COVID-19 outcomes despite booster vaccination. These data support the efforts to promote SARS-CoV-2 vaccination and increase the uptake of booster doses in order to prevent COVID-19-associated severe outcomes, particularly in older and frail individuals.

## Figures and Tables

**Figure 1 vaccines-12-01018-f001:**
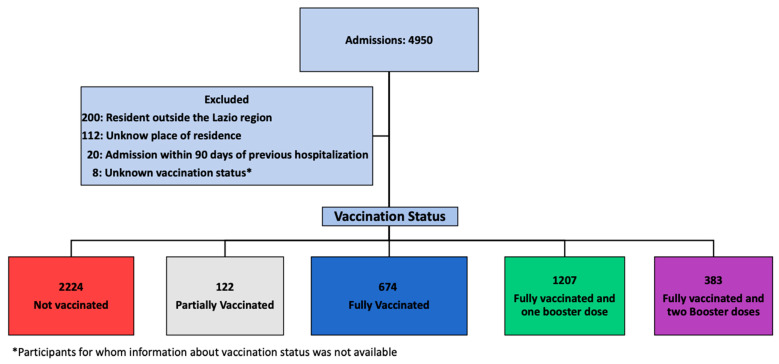
Study flow chart.

**Figure 2 vaccines-12-01018-f002:**
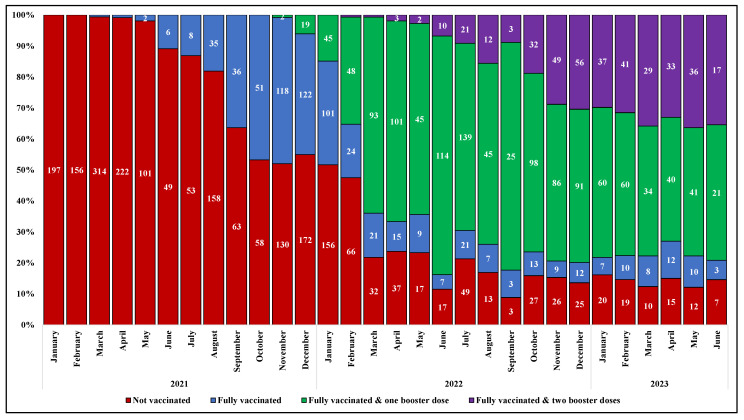
Prevalence of vaccination among study population by month and year of hospital admission.

**Figure 3 vaccines-12-01018-f003:**
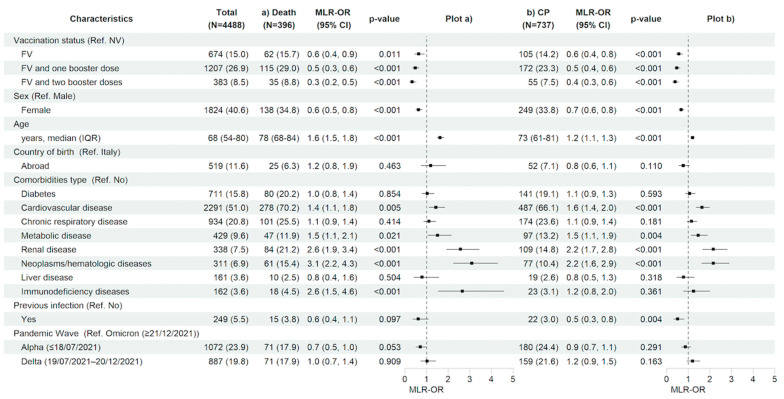
Characteristics of the participants and multivariable logistic regression results with forest plot representation for (**a**) in-hospital death and (**b**) clinical progression within 28 days from hospital admission, in the entire study population (n = 4488). For age, MLR-OR is for 10-year increase. Abbreviations: N, number of participants; MLR, multivariable logistic regression; OR, odds ratio; CP, clinical progression; CI, confidence interval; Ref, reference category; NV, not vaccinated; FV, fully vaccinated; IQR, interquartile range.

**Figure 4 vaccines-12-01018-f004:**
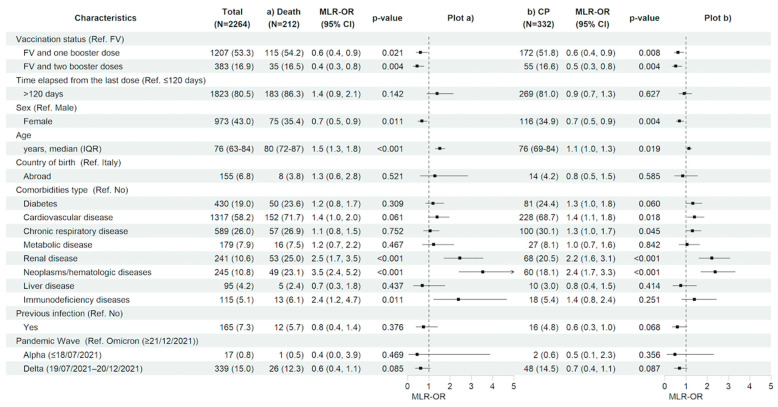
Characteristics of the participants and multivariable logistic regression results with forest plot representation for (**a**) in-hospital death and (**b**) clinical progression within 28 days from hospital admission, restricted to vaccinated study population (n = 2726). For age, MLR-OR is for 10-year increase. Abbreviations: N, number of participants; MLR, multivariable logistic regression; OR, odds ratio; CP, clinical progression; CI, confidence interval; Ref, reference category; FV, fully vaccinated; IQR, interquartile range.

**Table 1 vaccines-12-01018-t001:** Study participants’ characteristics at hospital admission according to the vaccination status (n = 4488).

Characteristics	Vaccination Status				Total (n = 4488)	*p*-Value
	Not Vaccinated n = 2224 (49.6%)	Fully Vaccinated (FV) n = 674 (15.0%)	FV & One Booster Dose n = 1207 (26.9%)	FV & Two Booster Doses n = 383 (8.5%)		
Females, n (%)	851 (38.3)	268 (40.9)	528 (43.2)	177 (45.6)	1824 (40.6)	0.005
Age (years), median (IQR)	59 (48–72)	68 (55–79)	76 (64–84)	82 (74–87)	68 (54–80)	<0.001
Age class (years), n (%)						<0.001
18–39	280 (12.6)	44 (6.5)	45 (3.7)	5 (1.3)	373 (8.3)	
40–59	870 (39.1)	176 (26.1)	192 (15.9)	18 (4.7)	1256 (28.0)	
60–79	776 (34.9)	294 (43.6)	503 (41.7)	139 (36.3)	1712 (38.2)	
≥80	298 (13.4)	160 (23.7)	468 (38.8)	221 (57.7)	1147 (25.6)	
Non-Italian born, n (%)	364 (16.4)	72 (10.7)	77 (6.4)	6 (1.6)	519 (11.6)	<0.001
Days from symptoms onset to hospital admission, median (IQR) (n = 3497)	6 (3–9)	4 (2–7)	2 (1–5)	2 (1–4)	4 (2–8)	<0.001
Comorbidities, n (%)						
Diabetes	281 (12.6)	135 (20.0)	200 (16.6)	95 (24.8)	711 (15.8)	<0.001
Cardiovascular disease	974 (43.8)	361 (53.6)	690 (57.2)	266 (69.5)	2291 (51.1)	<0.001
Chronic respiratory disease	345 (15.5)	146 (21.7)	321 (26.6)	122 (31.9)	934 (20.8)	<0.001
Metabolic disease	250 (11.2)	63 (9.4)	84 (7.0)	32 (8.4)	429 (9.6)	0.001
Renal disease	97 (4.4)	61 (9.1)	125 (10.4)	55 (14.4)	338 (7.5)	<0.001
Neoplasms/hematologic diseases	66 (3.0)	49 (7.3)	151 (12.5)	45 (11.8)	311 (6.9)	<0.001
Liver disease	66 (3.0)	30 (4.5)	49 (4.1)	16 (4.2)	161 (3.6)	0.166
Immunodeficiency diseases	47 (2.1)	28 (4.2)	65 (5.4)	22 (5.7)	162 (3.6)	<0.001
Number of comorbidities, n (%)						<0.001
0	859 (38.6)	164 (24.3)	230 (19.1)	46 (12.0)	1299 (28.9)	
1	806 (36.2)	240 (35.6)	470 (38.9)	131 (34.2)	1647 (36.7)	
2	395 (17.8)	190 (28.2)	342 (28.3)	121 (31.6)	1048 (23.4)	
≥3	164 (7.4)	80 (11.9)	165 (13.7)	85 (22.3)	494 (11.0)	
Previous infection *, n (%)	84 (2.8)	54 (8.0)	86 (7.1)	25 (6.5)	249 (5.6)	<0.001
Vaccine type (first dose), n (%)						<0.001
BNT162b2	Not applicable	447 (66.3)	945 (78.3)	313 (81.7)	1705 (75.3)	
mRNA-1273	Not applicable	70 (10.4)	147 (12.2)	52 (13.6)	269 (11.9)	
ChAdOx1	Not applicable	100 (14.8)	94 (7.9)	17 (4.4)	111 (9.3)	
Ad26.COV2.S	Not applicable	57 (8.5)	21 (1.7)	1 (0.3)	79 (3.5)	
Time elapsed from the last dose, n (%)						<0.001
≤120 days	Not applicable	113 (16.8)	220 (18.2)	108 (28.2)	441 (19.5)	
>120 days	Not applicable	561 (83.2)	987 (81.8)	275 (71.8)	1823 (80.5)	
Pandemic wave, n (%)						<0.001
Alpha (≤8 July 2021)	1055 (47.4)	17 (2.5)	0 (0)	0 (0)	1072 (23.9)	
Delta (19 July 2021–5 December 2021)	548 (24.6)	328 (48.7)	11 (0.9)	0 (0)	887 (19.8)	
Omicron (≥6 December 2021)	621 (27.9)	329 (48.8)	1196 (99.1)	383 (100	2529 (56.4)	
Laboratory markers of hyperinflammation, n (%)						
(n = 4421) Lymphocytes < 1, ×10^3^/μL	1242 (57.0)	347 (52.0)	577 (48.3)	193 (50.7)	2359 (53.4)	<0.001
(n = 4270) C-reactive protein > 3, mg/dL	1284 (61.2)	392 (61.7)	755 (64.9)	252 (67.2)	2683 (62.8)	0.049
(n = 3721) Ferritin > 500, ng/mL	896 (47.1)	182 (34.3)	254 (26.0)	62 (19.9)	1394 (37.5)	<0.001
Pneumonia, n (%)	2118 (95.2)	600 (89.0)	977 (80.9)	339 (88.5)	4034 (89.9)	<0.001

* If it occurred more than 90 days before the current diagnosis. Abbreviations: n, number of participants; IQR, interquartile range.

**Table 2 vaccines-12-01018-t002:** Main clinical outcomes by vaccination status.

Characteristics	Vaccination Status				Total Population	*p*-Value
	Not Vaccinated	Fully Vaccinated (FV)	FV & One Booster Dose	FV & Two Booster Doses		
	n = 2224	n = 674	n = 1207	n = 383	n = 4488	
Clinical outcomes						
Length of hospitalization (days), median (IQR)	14 (9–22)	14 (9–23)	14 (9–23)	14 (8–23)	14 (9–22)	0.437
Admission to ICU within 28 days from hospital admission, n (%)	364 (16.4)	79 (11.7)	122 (10.1)	39 (10.2)	604 (13.5)	<0.001
Length of ICU stay (days), median (IQR)	14 (8–27)	11 (6–36)	14 (5–27)	14 (5–25)	14 (7–27)	0.622
In-hospital death within 28 days from hospitalization, n (%)	184 (8.3)	62 (9.2)	115 (9.5)	35 (9.1)	396 (8.8)	0.627
Clinical progression within 28 days from hospitalization, n (%)	405 (18.2)	105 (15.6)	172 (14.3)	55 (14.4)	737 (16.4)	0.012
Virological outcomes						
(n = 4041) Viral shedding within 28 days from hospitalization, n (%)	1692 (83.2)	495 (82.4)	959 (89.2)	307 (92.2)	3453 (85.5)	<0.001

Abbreviations: n, number of participants; IQR, interquartile range; ICU, Intensive Care Unit.

## Data Availability

The data presented in this study are available upon reasonable request from the corresponding author.

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
