# Peer review of "Impact of Anti-SARS-CoV-2 Vaccination on Disease Severity and Clinical Outcomes of Individuals Hospitalized for COVID-19 Throughout Successive Pandemic Waves: Data from an Italian Reference Hospital"

_vaccines, 2024, doi:10.3390/vaccines12091018_

Round 1

Reviewer 1 Report

Comments and Suggestions for Authors

REVIEWER'S REPORT

Manucsript title: Impact of anti-SARS-CoV-2 vaccination on disease severity and clinical outcomes of individuals hospitalized for COVID-19 3 throughout successive pandemic waves: data from an Italian reference hospital.  (Authors: Mondi et al.)

The study described in this manuscript aimed to compare clinical characteristics and in-hospital outcomes of patients requiring hospitalisation for severe SARS-CoV-2 infection based on vaccination status between January 2021 and June 2023, coinciding with the launch of the anti-SARS-CoV-2 vaccine campaign in Italy.

  This manuscript is well-described and pertinent, in my opinion, but before it publishing I suggest a few brief remarks. Despite the fact that the article's text is quite clearly written and, I believe, understandable even to a reader working in a different scientific field, I recommend that some lines be corrected or rephrased to make it more clearer to the reader.

   To my mind, the following passages in the text need to be corrected:

In Abstract, the sentence in lines 26-27 must be corrected/rephrased.

In Introduction, the last part of the sentence (line 47) should be corrected.

The corrections/rephrasing are needed for sentences in Discussion (lines 318-324 ). 

 In Conclussion, the final part of sentence (line 371-372) should be written as "... particularly in older and frail individuals."

In my opinion, the subsection on Statistical analysis (page 3) should include additional references as well as a more thorough description of the statistical approaches used in this study should be provided.

Comments on the Quality of English Language

To my mind, the English language of the manuscript text just needs to be slightly corrected.

Reviewer 2 Report

Comments and Suggestions for Authors

In this manuscript entitled ‘Impact of anti-SARS-CoV-2 vaccination on disease severity and clinical outcomes of individuals hospitalized for COVID-19 throughout successive pandemic waves: data from an Italian reference hospital’, authors systematically analyzed the association between vaccination and disease severity and clinical outcomes of inpatients associated with COVID-19 in one hospital from Jan. 2021-Jun. 2023. The study is finely designed and well-written. Through fairly sound evidence and accurately stratified data analysis, authors reached a series of solid conclusion. This study is a good supplement to some previous similar studies (such as doi.org/10.1172/JCI167339, doi.org/10.1038/s41467-023-41537-7 and doi.org/10.1016/S2213-2600(23)00015-2); taken together, they contribute to the understanding of vaccination against severe COVID-19 and death in the real world which is now repeatedly experiencing various waves of variants. On the other hand, this study also verifies the correctness and soundness of some previous molecular studies (see doi.org/10.1038/s41586-022-04778-y, doi.org/10.1038/s41586-022-04466-x,doi.org/10.1016/j.cell.2022.04.009), which together indicate the effectiveness of booster-induced memory B cells against Omicron and its variants in atomical, cellular and animal levels. The authors are highly encouraged to discuss this in the manuscript to better explain the lifted protection against COVID-19 after vaccination observed here in clinical level. All in all, this manuscript provides profound and valuable insights into the control of COVID-19 for all the people, especially the vulnerable individuals. In addition, some minor issues also need to be corrected. 1) Please check if it is ‘contain’ or ‘control’ in line 47. 2) The reference format should be uniform and in accordance with the guidance of the journal.

Reviewer 3 Report

Comments and Suggestions for Authors

Comments and suggestions

The aim of the present study was to compare the clinical characteristics and hospital outcomes of

patients requiring hospitalization for severe SARS-CoV-2 infection according to vaccination status

over a 30-month observation period from the date of introduction of the

anti-SARS drug -CoV-2 vaccine campaign in Italy. The authors describe a topic widely reviewed and published in the medical literature, however a strength of the work is the 30-month follow-up.

Summary section:

1. Delete the first sentence of the summary section since the bibliographic search I performed showed a wide variety of studies on this topic.

2. Specify how many waves of COVID-19 were evaluated

3. Indicate which vaccine was used in each wave

4. Keywords: MeSH terms should be used

Introduction section:

5. Restructure the first sentence since the pandemic has passed.

6. Describe the types of vaccines available in each wave of the disease in Italy.

Materials and methods section:

7. Specify whether it was moderate or severe COVID-19.

8. Indicate the reason for the follow-up time in each group.

Results section:

9. In figure 1, it has not been defined or explained what is meant by unclear vaccination status among those excluded.

10. A graph representing the predictors of hospital death and clinical progression at 28 days would be better.

Conclusion section:

11. The conclusions need to be clearer and reflect the results based on the objectives of the study. Restructure

Comments on the Quality of English Language

Minor editing of English language required.
